# Production of a *Bacillus anthracis* Secretome with Suitable Characteristics as Antigen in a Complement Fixation Test

**DOI:** 10.3390/life12020312

**Published:** 2022-02-19

**Authors:** Domenico Galante, Viviana Manzulli, Adelia Donatiello, Antonio Fasanella, Barbara Chirullo, Massimiliano Francia, Valeria Rondinone, Luigina Serrecchia, Lorenzo Pace, Michela Iatarola, Michela Tarantino, Rosanna Adone

**Affiliations:** 1Istituto Zooprofilattico Sperimentale della Puglia e della Basilicata, Anthrax Reference Institute of Italy, Via Manfredonia 20, 71121 Foggia, Italy; viviana.manzulli@izspb.it (V.M.); adelia.donatiello@izspb.it (A.D.); antonio.fasanella@izspb.it (A.F.); valeria.rondinone@izspb.it (V.R.); luigina.serrecchia@izspb.it (L.S.); lorenzo.pace@izspb.it (L.P.); michela.iatarola@izspb.it (M.I.); 2Department of Food Safety, Nutrition and Veterinary Public Health, Istituto Superiore di Sanità, Viale Regina Elena 299, 00161 Rome, Italy; barbara.chirullo@iss.it (B.C.); massimiliano.francia@iss.it (M.F.); michela.tarantino@iss.it (M.T.); rosanna.adone@iss.it (R.A.)

**Keywords:** *Bacillus anthracis*, Complement Fixation Test, ELISA, secretome, vaccination

## Abstract

In this study, we cultured the *Bacillus anthracis* vaccine strain Sterne 34F2 in a medium containing EDTA, and we assessed the best conditions to inhibit the activity of zinc-dependent metalloproteases to obtain a secretome containing a high concentration of non-degraded PA (PA_83_), as evaluated by the SDS-PAGE analysis. Then, we used this secretome as the antigen in a Complement Fixation Test (CFT) to monitor the production of antibodies against PA_83_ in the sera of rabbits vaccinated with Sterne 34F2 and then infected with a *B. anthracis* virulent strain to evaluate the potency of the vaccine. The PAS-based CFT results were compared with those obtained by using a commercial ELISA kit. The two serological tests gave similar results in terms of specificity and sensitivity, as the kinetics of the antibodies production was very similar. The Sterne 34F2 vaccine induced an antibody response to PA_83_, whose titer was not inferior to 1:8 in PAS-based CFT and 42 kU/mL in PA_83_-based ELISA, respectively, in all vaccinated rabbits. Our opinion is that the PAS-based CFT can be successfully employed in humans and in animals for epidemiological retrospective studies or post-vaccination monitoring. We also suggest the use of our method to test the efficacy of veterinary anthrax vaccines.

## 1. Introduction

*Bacillus anthracis*, the causative agent of anthrax, produces a complex secretome whose composition varies widely, depending on different growth conditions [1]. The most characterized proteins produced by *B. anthracis* are the toxin components lethal factor (LF), protective antigen (PA) and edema factor (EF), secreted during the exponential phase of growth [2,3].

During anthrax infection, the host cell intoxication involves several steps: in the first step, the PA_83_ (protein of 83 kDa) binds to its cellular receptor and is cleaved by a furin-like protease, which removes a 20-kDa region. The remaining PA_63_ protein forms an oligomeric structure, which creates binding sites for EF or LF and promotes the translocation of the EF/LF complex into the cytosol. Following endocytosis, LF and EF exert their toxic effects: LF is a zinc-dependent metalloprotease that cleaves several mitogen-activated protein kinases, while EF is a calmodulin-dependent adenylate cyclase that causes local edema and impairs neutrophil function. The lethal toxin (LT) is composed by PA and LF, both of which are individually nontoxic [4,5,6].

The PA has a crucial role for immune inactivation of the anthrax toxins, since it induces the production of neutralizing antibodies, which are essential to mediate protection against anthrax [7,8,9]. A clear correlation has been demonstrated between neutralizing antibodies and protective immunity. It has been shown that the effectiveness of anthrax vaccines mainly depends on the magnitude of humoral response anti-PA induced in animals or humans [7,10,11,12]. In addition to toxin proteins, in media containing bicarbonate and providing a 5% CO_2,_ as occurs during the infection of mammalian tissues, *B. anthracis* secretes many other proteins that play an important role in the virulence process. Proteases and degradative enzymes have been identified that are able to inactivate host proteins, cleave antimicrobial host factors and modify secreted bacterial virulence factors, thus serving accessory functions for the full virulence of *B. anthracis* [13].

Two major proteases have been identified in supernatants of *B. anthracis* cultured in rich media: the neutral protease Npr599 and the zinc-dependent immune inhibitor A1 (InhA1), belonging to families M4 and M6 of the metalloproteases, respectively [14]. InhA1 is produced only by pathogenic members of the *Bacillus* genus [15].

These proteases are involved in the cleavage of host proteins and are able to modulate, directly and indirectly, the composition of *B*. *anthracis* secretome during the infection [15]. InhA1 regulates the level of toxin proteins PA, LF and EF. After the cultivation of *B. anthracis* in rich media, the level of toxin proteins remarkably decreased after the exponential growth phase but remained elevated when the *B. anthracis* strain, deleted of InhA1, was cultured under the same conditions [15]. InhA1 can degrade toxin proteins; PA was fully degraded following incubation with InhA1, resulting in fragments of different masses, even if the amino acid target sequence for cleavage was not specific [15].

In this study, we used the same proteomic approach to modulate the composition of *B. anthracis* secretome in order to avoid the degradation of toxin proteins. Particularly, our goal was to obtain a secretome containing a high concentration of non-degraded PA_83_ that could be used as a specific antigen in a Complement Fixation Test (CFT) for detecting antibodies to PA induced in humans or animals following vaccination or naturally acquired infection. The CFT is a very sensitive and specific test; it is easily standardized, not expensive and not species-specific [16].

To produce a suitable secretome, we cultured the attenuated vaccine strain *B. anthracis* Sterne 34F2 in a medium in which was added EDTA, a very potent zinc-chelating agent able to inhibit the activity of zinc-dependent metalloproteases, at different times of incubation. The effect of EDTA on the concentration of toxin proteins and on the degradation was evaluated in cultures by SDS-PAGE analysis.

We developed a protocol able to produce a secretome containing the highest concentration of non-degraded PA_83_.

Then, we evaluated the efficiency of this secretome, hereinafter named PAS, when used as an antigen in a PAS-based CFT in detecting antibodies to PA in serum samples collected from Sterne-vaccinated rabbits. The CFT results were compared with those obtained by testing the same samples by an ELISA test employing purified toxin PA_83_ as the coating antigen.

## 2. Materials and Methods

### 2.1. Bacterial Strain

The *B. anthracis* strain Sterne 34F2, used for the preparation of a routine veterinary vaccine, was supplied by the Anthrax Reference Institute of Italy (Ce.R.N.A.) of the Istituto Zooprofilattico Sperimentale of Puglia and Basilicata (Foggia, Italy). This institute is charged with the production of the Sterne 34F2 vaccine for veterinary use, routinely used for the immunization of susceptible animal species in Italy. The *B. anthracis* strain Sterne, developed by Max Sterne in 1937 [17], is an attenuated, toxigenic and unencapsulated strain characterized by a plasmid pattern pXO1^+^/pXO2^−^. On the pXO1 plasmid is present the genes encoding for toxins, while on the pXO2 plasmid is present the gene for the capsule.

### 2.2. Serum Samples

The serum samples used in this study were collected from thirty-five New Zealand White rabbits vaccinated with *B. anthracis* Sterne 34F2 and then infected with the virulent strain A0843 to evaluate the protective activity of the vaccine, according to the protocol followed by the Ce.R.N.A. for the control of the anthrax vaccine until 2015.

Experiments were conducted in accordance with the European Legislation (Directive 86/609/EEC) relating the welfare of animals used in scientific experiments.

Previously, we tested these samples to detect the presence of antibodies against *B. anthracis* in order to demonstrate the relationship between the humoral response and efficiency of the vaccine. The examined serum samples were collected at different times following vaccination. Briefly, 35 rabbits were vaccinated subcutaneously (s.c.) twice at a 15-day interval with a 1-mL dose, each containing 1.3 × 10^7^ live spores of Sterne 34F2 vaccine. Nine unvaccinated rabbits were kept as controls.

Blood samples were collected from all rabbits prior to vaccination (day 0). Fifteen days after the first vaccination, the rabbits were bled (15 dpv) and then were vaccinated again as described above; fifteen days after the second vaccination, all vaccinated rabbits were bled (30 dpv) and then were challenged with 200 LD_50_ of the *B. anthracis* virulent strain A0843 [18] to evaluate the protective activity of the vaccine; instead, the control group not vaccinated was challenged with 20 LD_50_ of the same virulent strain [19].

After the challenge, the unvaccinated rabbits died within 72–96 h, as expected, while all vaccinated rabbits survived. Fifteen days after the challenge, all survived rabbits were bled (45 dpv).

### 2.3. Production and Characterization of B. anthracis Sterne 34F2 Secretome

*B. anthracis* Sterne 34F2 strain was seeded on blood agar plates and incubated at 37 °C for 24 h. After incubation, the bacterial growth was collected from each plate (1.73 × 10^8^ CFU) and transferred to five bottles, each containing 100 mL of RPMI-1640 medium (Roswell Park Memorial Institute 1640) and incubated at 37 °C in atmosphere with 5% CO_2_. The RPMI-1640 contains amino acids, vitamins and inorganic salts, and it lacks proteins, lipids or growth factors; this medium is a sodium bicarbonate buffer system (2 g/L) and, therefore, requires a 5–10% CO_2_ environment to maintain the physiological pH.

To inhibit the activity of zinc-dependent proteases, 1 mL of the EDTA (ethylenediaminetetraacetic acid) 10-μM solution was added to each culture at different incubation times: at the beginning of incubation (T0), one hour (T1), two hours (T2) and three hours (T3) later. A culture without EDTA (CTRL) was used as the control.

After 4-h incubation, each suspension was centrifuged at 6.026× *g* for 10 min at 4 °C, and the supernatant was filtered with 0.80- and then with 0.45-µm filters. The filtrate was divided into centrifugal devices 10 K with a Molecular Weight Cut-off (MWCO) of 30–90 kDa (Pall Life Sciences, Ann Arbor, MI, USA), centrifuged at 3.075× *g* for 30 min at 4°C and then at 1.107× *g* for 2 min at 4 °C.

The total concentration of proteins of each filtrate was quantified by the Bradford assay, and the optical density (OD) of the samples was assessed by using a NanoDrop 1000 instrument (NanoDrop1000 Spectrophotometer, Thermo Fisher Scientific, Waltham, MA, USA).

Samples were diluted 1:4 in Bradford reagent, and the OD was measured at 595 nm.

To evaluate the effect of the EDTA on protein composition and degradation, a qualitative analysis of the protein extracts was performed by using SDS-PAGE (polyacrylamide gel electrophoresis with sodium dodecyl sulfate), which allows the separation of proteins based on their molecular weight. Briefly, 900 µL of 4XLaemmli sample buffer solution (Bio-Rad Laboratories, Hercules, CA, USA) were added to 100 µL of 2-mercaptoethanol. Fifteen microliters of solution, previously prepared, and 45 μL of sample were dispensed in 1.5-mL tubes. Subsequently, each sample was incubated at 100 °C for 5 min.

Later, 40 μL of each sample and 10 μL of Precision Plus Protein™ All Blue Prestained Protein Standards (Bio-Rad Laboratories, Hercules, CA, USA) were loaded separately into the wells of 4–20% Criterion™ TGX™ Precast Midi Protein Gel and 4–20% precast polyacrylamide gel (Bio-Rad Laboratories, Hercules, CA, USA), previously placed in the appropriate electrophoretic cell (Criterion™ Vertical Electrophoresis Cell, Bio-Rad, Hercules, CA, USA).

Electrophoresis was performed at room temperature for 55 min using a constant voltage of 180 V. At the end of the run, the gel was immersed in a Blue Brillant Coomassie solution for 1 h at room temperature with gentle agitation for the staining of proteins. Finally, the gel was washed with a destaining solution.

### 2.4. Assessment of a Complement Fixation Test Using the Most Suitable B. anthracis Secretome (PAS) as Antigen

The CFT is a sensitive and specific serological test, widely used for the diagnosis of many infectious diseases—in particular, for brucellosis. In this test, the presence of an antigen–antibody complex, which fixes the complement, is made visible by an indicator system consisting of sheep erythrocytes coated with anti-sheep erythrocytes antibody (hemolysin). For this study, erythrocytes, hemolysin and the complement were supplied by Emozoo Snc of Ripabelli G. & C. (Siena, Italy). The calcium magnesium veronal buffer, pH 7.2 ± 0.1 (Lonza, Walkersville, Inc., Walkersville, MD, USA), added with 0.1% of ovine serum albumin (VBA), was used as the diluent.

We assessed a CFT in which the secretome produced by the *B. anthracis* culture, showing the highest protein concentration and the lowest degradation of PA_83_, as described above, was used as the antigen. The secretome with these suitable characteristics (named PAS) was then titrated as prescribed by the OIE Manual of Diagnostic Tests for the diagnosis of brucellosis [20] to identify the PAS dilution providing the best sensitivity and specificity when used as the antigen in CFT.

For this purpose, two-fold dilutions of PAS were tested in CFT against positive and negative control sera provided by the Ce.R.N.A., as previously described [16,21].

The positive control serum was a pool of sera collected from experimentally anthrax-infected rabbits, while the negative control was a pool of sera from healthy, unvaccinated rabbits.

### 2.5. Elisa Assay

Since ELISA is considered the most effective serological method to evaluate the humoral response induced by *B. anthracis*, all rabbit sera tested by the PAS-based CFT were evaluated in parallel with the commercial Rabbit Anti-Anthrax Protective Antigen (PA_83_) ELISA kit (Alpha Diagnostic International, San Antonio, TX, USA) used for the detection and quantification of the anti-PA IgG in rabbit sera. In this kit, the purified recombinant protein PA_83_ is immobilized as the antigen in microtiter wells. As prescribed, rabbit antisera were diluted at 1:2000, identified as the best dilution and 100 µL of each dilution were added to the wells. Five standards provided by the kit containing antibodies calibrated in anti-PA_83_ activity units (U/mL) from 10 to 160 U/mL were also tested. After incubation, the plates were washed, and the anti-rabbit IgG-HRP conjugate was added to each well. After incubation and washing, the chromogenic substrate TMB was added; when an optimal contrast was reached, the coloration was stopped by adding sulfuric acid solution, which enables the accurate measurement of the color intensity. The absorbance was measured at 450 nm using an ELISA microtiter plate reader. The concentration of IgG anti-PA_83_ detected in the serum samples was calculated from the curve of standards; each final titer, considering the dilution factor, was expressed in kU/mL.

### 2.6. Statistical Analysis of PAS-Based CFT and ELISA Results

The antibody titer of Sterne-vaccinated rabbits, detected by PAS-based CFT and ELISA, were analyzed at different time points (0, 15, 30 and 45 dpv) by the Kruskal–Wallis multiple comparison one-way ANOVA test. Significant differences between bars were marked with different letters. *p*-values of 0.05 or less were considered significant. These symbols were used to indicate the statistical significance: * *p* < 0.05, ** *p* < 0.01, *** *p* < 0.001 and **** *p* < 0.0001.

## 3. Results

### 3.1. SDS-PAGE Profiles of B. anthracis Sterne 34F2 Secretomes

The composition and integrity of the toxin proteins produced in *B. anthracis* cultures after the addition of EDTA at different times of incubation were evaluated by SDS-PAGE analysis. The stained protein patterns of cultures from T0 to T3 are shown in Figure 1. The components ranged in molecular weights from about 20 to 250 kDa. However, the addition of the 10-µM EDTA solution strongly altered the integrity and the concentration of the toxin proteins: the lowest degradation of PA_83_ was obtained when EDTA was added one hour after the beginning of the incubation (Figure 1, lane 3). In contrast, the higher degradation of PA_83_ occurred when EDTA was added at the beginning of the incubation period (lane 2), but it was inferior two and three hours later (lanes 4 and 5, respectively). A culture without EDTA, used as the negative control (lane 6), showed the almost complete degradation of PA_83_ but with the presence of two marked bands of 63 kDa and approximately 50 kDa.

Cultures at different times of incubation with EDTA were analyzed all together by SDS-PAGE. The secretome after 1 h of incubation (T1) with EDTA 10-µM solution, thanks to its better quality (minor degradation), was suitable as the PAS antigen and could be used in the PAS-based CFT.

### 3.2. Use of the Purified Secretome (PAS) as Antigen in CFT

The PAS, as described above, was tested in a block titration against the positive antiserum and negative serum to determine the optimal concentration for its use as the antigen in CFT. The most suitable dilution was proven to be 1:4, corresponding to a protein concentration of 150 µg/mL. At this concentration, the antigen PAS was able to react with the positive control, providing the expected titer, while it did not react with the negative control serum.

### 3.3. PAS-Based CFT and ELISA Results

The results of the PAS-based CFT and ELISA performed on the sera of Sterne-vaccinated rabbits are shown in Figure 2A,B. As shown, all serum samples from the rabbits bled before vaccination and from the unvaccinated controls were seronegative, indicating 100% specificity of both tests.

The PAS-based CFT results confirmed that the titer 1:2 should be considered the reactivity threshold of the reaction, as indicated in our previous study [16] at 15 dpv; all vaccinated rabbits reacted weakly, the majority of them giving a titer of 1:4. A strong increase of the seroconversion was detected in all rabbits at 30 dpv (15 days after the second vaccination), when the prevalence of the CFT titers ranged from 1:16 to 1:32 (Figure 2A).

It is interesting to notice that, at 45 dpv, 15 days after the challenge with the *B. anthracis* virulent strain, in sera collected from vaccinated rabbits, the CFT titers to PAS did not increase, as expected, but slightly decreased (Figure 2A). The rabbits used as controls were not tested at this time, because they died 72–96 h after the challenge.

The results of the ELISA confirmed the presence of specific IgG anti-anthrax PA_83_ in all the vaccinated rabbits; the kinetics produced were very similar to those observed in PAS-based CFT (Figure 2B). Fifteen days post-vaccination with Sterne, in all samples examined, the measure of OD_450_ was over the reactivity threshold value of 0.05. However, the reaction was still weak, titers ranging from 1.8 to 9.6 kU/mL (mean value: 6.1 kU/mL). At 30 dpv, the IgG production strongly increased in all the vaccinated rabbits, IgG titers ranging from 42 to 210 kU/mL with a mean value of 106.12 kU/mL (Figure 2B).

Finally, in contrast with the PAS-based CFT results, when vaccinated rabbits were tested at 45 dpv, 15 days after the challenge, the IgG titers further increased, reaching the mean value of 172.7 kU/mL (Figure 2B).

## 4. Discussion

Bacteria are able to modulate the environment to promote their survival and proliferation through the secretion of proteases that cleave extracellular substrates [15,22,23,24]. In pathogenic bacteria, secreted proteases inactivate essential host proteins, cleave antimicrobial host factors and modify other secreted bacterial virulence factors [13,15,25]. The *B. anthracis* secretome includes the protective antigen,the lethal factor and the edema factor, which are the components of anthrax toxins and other proteins with known or potential roles in anthrax disease; the proteolytic action of secreted metalloproteases may drastically alter the composition of the *B. anthracis* secretome, limiting the integrity and the concentration of the toxin proteins [15].

In order to obtain a *B. anthracis* secretome containing nondegraded toxin proteins, evaluated as a potential diagnostic antigen, we cultured the vaccine strain *B. anthracis* Sterne 34F2 in RPMI-1640 medium containing the zinc-chelating agent EDTA, which is able to inhibit the activity of zinc-dependent metalloproteases. EDTA was added at different times of incubation, and its effect on the toxin protein concentration and degradation was evaluated by SDS-PAGE analysis.

We identified the best protocol that allowed the production of the most suitable secretome (PAS): based on our experiments, 1 mL of 10-µM EDTA must be added to the *B. anthracis* culture one hour after the beginning of the incubation in RPMI to obtain the highest concentration of nondegraded PA_83_, as demonstrated by SDS-PAGE analysis (Figure 1).

After determining the optimal culture conditions to obtain nondegraded PA and subsequent optimal dilutions, PAS was used as antigen in a CFT, and the ability of the PAS-based CFT to detect antibodies directed at PA_83_ was evaluated.

For this purpose, serum samples collected from Sterne-vaccinated and from unvaccinated rabbits, both subjected to infection with a virulent *B. anthracis* strain, were collected at different sampling times during the procedures prescribed to evaluate the potency of the Sterne 34F2 vaccine. All vaccinated rabbits survived the challenge, indicating that the antibody response induced by the vaccine was protective, while all controls died within 72–96 h after infection, as expected.

All serum samples were tested with our PAS-based CFT and, in parallel, with a diagnostic ELISA kit routinely used for the detection and quantification of the anti-PA IgG in rabbit sera, which utilizes the purified recombinant protein PA_83_ as a coated antigen.

As shown (Figure 2A,B), the two serological tests gave similar results in terms of specificity and sensitivity. In fact, all unvaccinated control rabbits and rabbits bled prior to vaccination were seronegative when tested by PAS-based CFT and ELISA, indicating 100% specificity of both tests. Moreover, the kinetics of antibody production were very similar; all vaccinated rabbits presented low IgG titer at 15 dpv, while the IgG titers peaked at 30 dpv.

The only difference between the two methods refers to the anti-PA_83_ titers measured at 45 dpv, fifteen days after the challenge with the *B. anthracis* virulent strain. While the PAS-based CFT showed no titer increase, the IgG titer measured by ELISA showed a further weak increase compared to the previous 30 dpv sampling period.

This should not be surprising, since different serological tests detect antibodies with distinct isotopes and behavior characteristics, such as the ability to fix the complement. Probably, the challenge with the virulent strain mainly stimulated antibodies that did not fix the complement.

It has already been shown that antibodies to PA are essential to mediate protection against anthrax and that the efficiency of anthrax vaccines mainly depends on their ability to elicit antibodies directed at this toxin component.

Our results indicated that rabbits showing a CFT antibody titer to PA_83_ ranging from 1:16 to 1:32, obtained after two vaccinations with Sterne 34F2, were protected against the challenge with the *B. anthracis* virulent strain. By the ELISA test, the protective titers to PA_83_ were over the threshold value of 42 kU/mL.

Based on these data, we could establish that a batch of vaccine is effective if it induces, in all experimental rabbits, an antibody response to PA_83_, whose titer is not inferior to 1:8 or 42 kU/mL in PAS-based CFT or PA_83_-based ELISA, respectively.

However, the CFT offers some benefits compared to ELISA; it is cheaper, may be easily standardized and is not species-specific, so it can be used to test human serum samples, as well as animal species, with the same reagents [16].

The assessment of this protocol to produce the PAS, consisting of nondegraded PA_83_ at a high concentration, allowed us to develop a PAS-based CFT that is able to detect and quantify the antibodies directed to PA_83_ with a very similar efficiency as the PA_83_-based ELISA.

## 5. Conclusions

In our opinion, this PAS-based CFT could be used to quantify the antibody response to PA_83_ following anthrax infection in humans and in animals to achieve epidemiological information or to confirm anthrax circulation in areas where the disease is endemic. In addition, it could be used to evaluate the protective antibody response following vaccination in humans (i.e., for army soldiers or healthcare operators).

We suggest the use of the PAS-based CFT to measure the potency of veterinary anthrax vaccines in replacement of the experimental infection with *B. anthracis* virulent strains, a practice carried out up to some years ago but now forbidden because of new rules concerning animal welfare. In fact, this practice caused considerable suffering to animals and exposed laboratory technicians to biosafety risks. Furthermore, the challenge with anthrax virulent strains requires the possession of biosafety level laboratories (BSL-3). Other experiments are in progress in our laboratory to evaluate the immunogenic characteristics of the purified PAS, including the evaluation of a possible use as a candidate vaccine against anthrax.

## Figures and Tables

**Figure 1 life-12-00312-f001:**
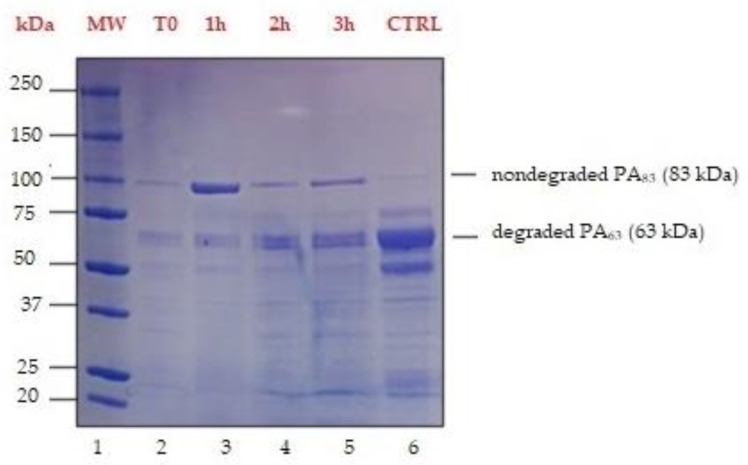
SDS-PAGE analysis of *B. anthracis* Sterne 34F2 secretomes. Protein molecular weight marker (lane 1). One milliliter of 10-μM EDTA solution was added to the Sterne 34F2 culture in CO_2_/RPMI (toxin-inducing condition) at different incubation times: at the beginning of the incubation period (lane 2—T0), one hour later (lane 3—T1), two hours later (lane 4—T2) and three hours later (lane 5—T3). A culture without EDTA was left as the negative control (lane 6—negative CTRL).

**Figure 2 life-12-00312-f002:**
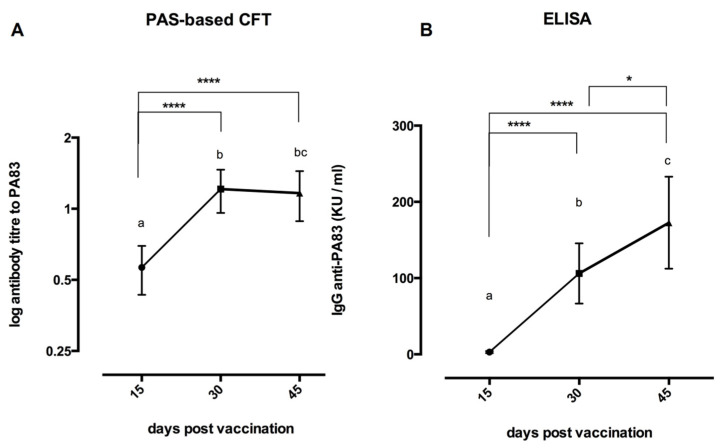
Antibody response following Sterne vaccination of thirty-five rabbits, measured by the PAS-based CFT (**A**) and by the Rabbit Anti-Anthrax Protective Antigen (PA_83_) ELISA Kit (**B**) at different days post-vaccination (dpv). Rabbits were vaccinated two times: at time 0 and at 15 dpv, with 1.3 × 10^7^ live spores of Sterne 34F2/rabbit. At 30 dpv, all rabbits were challenged with a *B. anthracis* virulent strain (200 LD_50_) and then were bled at 45 dpv. The titer of each sample was the highest dilution showing a positive reaction; the results were expressed as the log mean ± standard deviations and analyzed by the Kruskal–Wallis test multiple comparison one-way ANOVA. Significant differences between bars were marked with different letters. These symbols were used to indicate the statistical significance: * *p* < 0.05 and **** *p* < 0.0001.

## Data Availability

The data presented in this study are available on request from the corresponding author.

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
