# Peer review of "Production of a Bacillus anthracis Secretome with Suitable Characteristics as Antigen in a Complement Fixation Test"

_life, 2022, doi:10.3390/life12020312_

Round 1

Reviewer 1 Report

The manuscript presented by Galante and colleagues describes the development of a procedure to produce a B. anthracis (BA) 34F2 vaccine strain secretome in conditions to limit zinc-protease activity and avoid degradation of protective antigen (PA or PA83). The resulting PA83 secretome was then used to develop a CFT assay as a mean to monitor antibodies produced against PA83 in rabbits vaccinated against BA, with comparison of their results to a commercial ELISA procedure, with results showing that the CFT represent a suitable assay to epidemiological and vaccination monitoring purposes.

The manuscript is overall very well written, and reads very well as well, with each section very well crafted. To my opinion, this manuscript is ready for publication with a few very minor modifications.

Line 64. Provide a space in “InhA1 is”.

Line 149. “by using the SDS-PAGE”: delete “the”.

In line 210, you mention in the text that P values of 0.05 or less are considered significant, but then write in line 211: “*P<0.1”. I feel this is a typo and it should read *P<0.05. In relation to this statement, please make sure that Fig. 2 is appropriately labeled: does the * on the B panel means NS or that the data is significant with a P<0.05. Also check and modify if necessary the Fig. 2 legend (line 290).

For Fig. 1, I have a few suggestions so that the figure is a little easier to read. On the top section, maybe between the numbers and the picture, you could add a line that provides information about the EDTA treatment. Under 1, you could add MW; 2: T0; 3: 1h; 4: 2h, 5: 3h; 6: control. Also on the right side of the picture, arrows could point to the non-degrated PA83 and the degraded forms (PA63? PA50?). The idea is to have sufficient information on the figure to quickly get to the results independent of the text and legend.

I have a few suggestions on the Discussion section.

Line 294: replace though by through.

Line 302 (and 351): replace “not degraded” by “non-degraded”.

Line 310: replace later by after; line 311: add (Fig. 1) at the end of the sentence.

Line 312-313: the sentence feels weird. Suggestion: “After determining the optimal culture conditions to obtain non-degraded PA and subsequent optimal dilutions determined, PAS was used as…”.

Line 326: replace “resulted” by “were”.

Line 328: “in the two tests…” replace to “ all vaccinated rabbits presented low IgG titer at 15 dpv, while…

Line 399: replace “so much that” by “and that”.

Line 366: “exposes laboratory technicians to biosafety risks”.

The paper is overall a great read and very well presented, and we feel the data presented could provide critical insights into developing an effective yet affordable assay to monitor the efficacy of vaccination program in animals and human patients alike.

Author Response

Dear Reviewer,

thank you for your valuable comments and for helping us to improve our manuscript.

The manuscript presented by Galante and colleagues describes the development of a procedure to produce a B. anthracis (BA) 34F2 vaccine strain secretome in conditions to limit zinc-protease activity and avoid degradation of protective antigen (PA or PA83). The resulting PA83 secretome was then used to develop a CFT assay as a mean to monitor antibodies produced against PA83 in rabbits vaccinated against BA, with comparison of their results to a commercial ELISA procedure, with results showing that the CFT represent a suitable assay to epidemiological and vaccination monitoring purposes.

The manuscript is overall very well written, and reads very well as well, with each section very well crafted. To my opinion, this manuscript is ready for publication with a few very minor modifications.

Line 64. Provide a space in “InhA1 is”.

Done

Line 149. “by using the SDS-PAGE”: delete “the”.

Deleted “the” at the line 149

In line 210, you mention in the text that P values of 0.05 or less are considered significant, but then write in line 211: “*P<0.1”. I feel this is a typo and it should read *P<0.05. In relation to this statement, please make sure that Fig. 2 is appropriately labeled: does the * on the B panel means NS or that the data is significant with a P<0.05. Also check and modify if necessary the Fig. 2 legend (line 290).

It was a typo. I replaced “P<0.1” with “P<0.05” at the lines 210 and 290.

For Fig. 1, I have a few suggestions so that the figure is a little easier to read. On the top section, maybe between the numbers and the picture, you could add a line that provides information about the EDTA treatment. Under 1, you could add MW; 2: T0; 3: 1h; 4: 2h, 5: 3h; 6: control. Also, on the right side of the picture, arrows could point to the non-degraded PA83 and the degraded forms (PA63? PA50?). The idea is to have sufficient information on the figure to quickly get to the results independent of the text and legend.

Edited the Fig.1 as suggested

I have a few suggestions on the Discussion section.

Line 294: replace though by through.

Replaced "though" with "through"

Line 302 (and 351): replace “not degraded” by “non-degraded”.

Replaced “not degraded” with “non-degraded” throughout the text

Line 310: replace later by after;

Replaced "later" with "after" at the line 310

line 311: add (Fig. 1) at the end of the sentence.

Added (Fig. 1) at the end of the sentence at the line 311

Line 312-313: the sentence feels weird. Suggestion: “After determining the optimal culture conditions to obtain non-degraded PA and subsequent optimal dilutions determined, PAS was used as…”.

Reworded the sentence as suggested at the lines 312-313

Line 326: replace “resulted” by “were”.

Replaced “resulted” with “were”

Line 328: “in the two tests…” replace to “all vaccinated rabbits presented low IgG titer at 15 dpv, while…

Replaced “in the two tests…” with “all vaccinated rabbits presented low IgG titer at 15 dpv, while…"

Line 399: replace “so much that” by “and that”.

Replaced “so much that” with “and that”

Line 366: “exposes laboratory technicians to biosafety risks”.

Replaced "exposes laboratory technicians to many risks" with “exposes laboratory technicians to biosafety risks"

The paper is overall a great read and very well presented, and we feel the data presented could provide critical insights into developing an effective yet affordable assay to monitor the efficacy of vaccination program in animals and human patients alike.

Reviewer 2 Report

In this study, authors evaluated a complement fixation test (CFT) using a secretome containing a high concentration of not degraded PA83 as a specific antigen. The research design is meaningful. However, several flaws were found in the manuscript, which is required a major revision.

In Figure 1, lane 6, it looks like there is a band around lining with lanes 2-5, how to verify the band shown is the target band.

As shown (Figs. 2A and 2B), the two serological tests gave similar results in terms of specificity and sensitivity. The results were shown in a similar trend at 45 dpv. Additional time points can be added to find the appropriate testing time point.

In addition, no potential test was performed in added human serum samples.

Minors:

Complement Fixation Test (CFT) abbreviation should be listed in the abstract.

Line 64, InhA1is, correct it to InhA1 is.

Line 68-71, grammar check is needed.

100μl > 100 μl.

P values of 0.05 or less were considered significant. Therefore, no need to label 0.05 < p < 0.1.

Author Response

Dear Reviewer,  

thank you for your valuable comments and for helping us to improve our manuscript.

In this study, authors evaluated a complement fixation test (CFT) using a secretome containing a high concentration of not degraded PA83 as a specific antigen. The research design is meaningful. However, several flaws were found in the manuscript, which is required a major revision.

In Figure 1, lane 6, it looks like there is a band around lining with lanes 2-5, how to verify the band shown is the target band.

Thanks, this is a right observation. In the control lane (lane 6) a very light band is visible, because however there is the presence of non-degraded PA83, that is part of the protein composition of the secretome. The PA83 band of the secretome without EDTA is visibly reduced, it almost disappears (as now better described in the text of manuscript), if compared to the bands of secretome in addition with EDTA (lanes 2-5). The molecular weight of this band is the same of the others in the lane 2-5 as showed in Fig. 1.

As shown (Figs. 2A and 2B), the two serological tests gave similar results in terms of specificity and sensitivity. The results were shown in a similar trend at 45 dpv. Additional time points can be added to find the appropriate testing time point.

Thank you for your suggestion. Unfortunately, at the moment we don’t have other time points we can add at this study, but we will check other times in the next experiments.

In addition, no potential test was performed in added human serum samples.

Thank you for your considerations. We are a veterinary institute and usually we don’t receive human samples. However, in the future we can consider to make collaborations with some hospitals or public health institutes in order to test also human samples by our method.

Minors:

Complement Fixation Test (CFT) abbreviation should be listed in the abstract.

Added Complement Fixation Test abbreviation (CFT) in the abstract

Line 64, InhA1is, correct it to InhA1 is.

Added a space between InhA1 and is at the Line 64

Line 68-71, grammar check is needed.

Corrected the sentence at the lines 68-71

100μl > 100 μl.

Added a space between 100 and μl

P values of 0.05 or less were considered significant. Therefore, no need to label 0.05 < p < 0.1.

It was a typo. I replaced “P<0.1” with “P<0.05” at the lines 210 and 290

Round 2

Reviewer 2 Report

The authors answer the questions and revised the manuscript. No further comments.